# Asymmetric total synthesis of yuzurimine-type *Daphniphyllum* alkaloid (+)-caldaphnidine J

Lian-Dong Guo [1], Yan Zhang [1,2], Jingping Hu [1,2], Chengqing Ning [1], Heyifei Fu[1], Yuye Chen [1] & Jing Xu [1✉]

Ever since Hirata's report of yuzurimine in 1966, nearly fifty yuzurimine-type alkaloids have been isolated, which formed the largest subfamily of the *Daphniphyllum* alkaloids. Despite extensive synthetic studies towards this synthetically challenging and biologically intriguing family, no total synthesis of any yuzurimine-type alkaloids has been achieved to date. Here, the first enantioselective total synthesis of (+)-caldaphnidine J, a highly complex yuzurimine-type *Daphniphyllum* alkaloid, is described. Key transformations of this approach include a highly regioselective Pd-catalyzed hydroformylation, a samarium(II)-mediated pinacol coupling, and a one-pot Swern oxidation/ketene dithioacetal Prins reaction. Our approach paves the way for the synthesis of other yuzurimine-type alkaloids and related natural products.

[1] Shenzhen Grubbs Institute and Department of Chemistry and Guangdong Provincial Key Laboratory of Catalysis, Southern University of Science and Technology, 518055 Shenzhen, China. [2] These authors contributed equally: Yan Zhang, Jingping Hu. ✉email: xuj@sustech.edu.cn

The plant family of genus *Daphniphyllum* has produced a wide range of complex caged natural products—the *Daphniphyllum* alkaloids[1–4]. Owing to their challenging chemical structures and interesting biological profiles (such as anticarcinogenic, neurotrophic, and anti-HIV activity[5,6], these alkaloids have drawn much attention from the synthetic community[1,3,4]. Depending on the taxonomy, the diversified structures of the *Daphniphyllum* alkaloids can be categorized into 13–35 subfamilies[1,4,7]. To date, about 20 elegant total syntheses of the *Daphniphyllum* alkaloids, from seven subfamilies, have been reported by the research groups of Heathcock[8–15], Carreira[16], Li[17–20], Smith[21,22], Hanessian[23], Fukuyama/Yokoshima[24], Zhai[25], Dixon[26], Qiu[27], Gao[28], and Sarpong[29,30] groups, respectively. Moreover, our group has recently accomplished the asymmetric total syntheses of himalensine A[31], dapholdhamine B[32], and caldaphnidine O[33]. However, despite extensive synthetic studies[34–48], total synthesis of yuzurimine-type alkaloids—the largest subfamily of the *Daphniphyllum* alkaloids—has not been achieved so far.

Since the milestone achievement by Hirata in 1966[49], nearly 50 yuzurimine-type alkaloids were isolated, which account for almost one-sixth of all known *Daphniphyllum* alkaloids to date. It is recognized that the members of this subfamily share a unique, highly complex and caged hexacyclic skeleton (Fig. 1). Caldaphnidine J, our target molecule, was isolated by the Yue group in 2008[50]. It possesses a hexacyclic scaffold with six contiguous stereogenic centers, two quaternary centers, and an α,β,γ,δ-unsaturated carboxylic ester thus signifying a formidable synthetic challenge. Following our long-lasting interest in the synthesis of the *Daphniphyllum* alkaloids[31–33,51,52], we now wish to report an asymmetric total synthesis of yuzurimine-type alkaloid, (+)-caldaphnidine J. Our approach is featured with a Pd-catalyzed regioselective hydroformylation that furnishes the critical aldehyde motif, and the construction of the 5/5 bicyclic system using a SmI$_2$-promoted pinacol coupling and a one-pot Swern oxidation/ketene dithioacetal Prins reaction. Other notable attempts toward the 7/5 bicycle in our target molecule are also discussed.

## Results

**Retrosynthetic analysis.** As shown in Fig. 2, our retrosynthetic analysis indicates that (+)-caldaphnidine J could be converted from the intermediate **1** via Ni(0)-mediated reductive coupling reaction[53]. It was further envisioned that a ketene dithioacetal Prins reaction[54–57] would form the cyclopentene moiety in **1**, while compound **2** might be produced from the enyne acetal **3** via a carbocyclization cascade[58]. Lastly, compound **3** could be synthesized via a facile ring cleavage from the readily available chiral building block, tricyclic ketone **4**[33].

**Initial synthetic studies toward the 7/5 ring system.** The first stage of our investigation focused on the critical 7/5 ring scaffold assembly (Fig. 3). The treatment of the known tricycle **4**[33] with MeMgBr in the presence of CeCl$_3$ was followed by an oxidative cleavage of the intermediate diol producing dicarbonyl **5**. Subsequently, **5** was subjected to Wittig conditions to yield the enol methyl ether, which was then transformed into acetal **6**. Following Stang's protocol[59], ketone motif in **6** was converted into an enol triflate, which was then treated with pyridine affording alkyne **7**. Subsequent Sonogashira coupling yielded enyne acetal **3**. Inspired by Saá's impressive Nazarov carbocyclization cascade[58], enyne acetal **3** was treated with Brønsted acids such as TFA or HBF$_4$•OEt$_2$. However, none of the desired hydroazulenone **8** was detected. Alternatively, attempting to use a formic acid-promoted cyclization[60,61] between the thioalkyne and

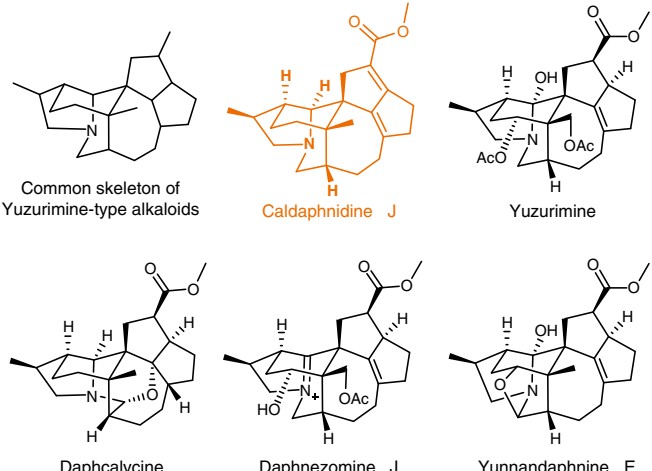

**Fig. 1 Yuzurimine-type *Daphniphyllum* alkaloids.** Our target molecule, caldaphnidine J, are highlighted in color, which posed a formidable synthetic challenge for accessing its total synthesis.

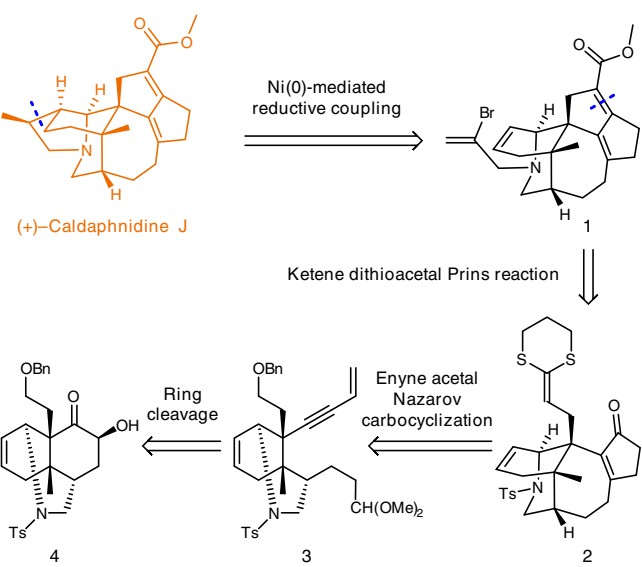

**Fig. 2 Retrosynthetic analysis of (+)-caldaphnidine J.** The key disconnections include a Ni(0)-mediated reductive coupling reaction, a ketene dithioacetal Prins reaction, an enyne acetal Nazarov carbocyclization and a ring cleavage strategy.

the acetal motifs in compound **9** returned only messy mixtures. It was postulated that the desired cyclization was prevented by the steric bulk at C-8 (caldaphnidine J numbering).

On the other hand, aldol condensation of **5** yielded solely cyclopentene **11**, but not cycloheptenone **12** possessing the desired seven-membered ring (Fig. 3). Instead, selective reduction of the aldehyde motif in **5** followed by an Appel reaction afforded the corresponding alkyl iodide, which then underwent an intramolecular alkylation upon treatment with LDA yielding the desired tricycle **13** in 69% overall yield. Inspired by Smith's impressive work[21,22], it was envisioned that a carbonylative Stille coupling reaction followed by a Nazarov cyclization would elaborate the desired bicycle **8**. While ketone **13** is quite similar to Smith's substrates[21,22], in our hands all of the deprotonation attempts failed resulting in no reaction at negative and decomposition events at elevated temperatures. It is quite possible that the corresponding enol triflate **14** is unstable by introducing increased ring strain to the 6/6/7 tricyclic system.

**Fig. 3 Initial studies towards the 7/5 ring system.** Bn benzyl, KHMDS potassium bis(trimethylsilyl)amide, LDA lithium diisopropylamide, Tf trifluoromethanesulfonyl, TFA trifluoroacetic acid, TEA triethylamine, Ts toluenesulfonyl.

**Total synthesis of (+)-caldaphnidine J.** The fruitless synthetic attempts toward the 7/5 bicyclic ring scaffold forced us to use the similar strategy to the total synthesis of caldaphnidine O[33]. As depicted in Fig. 4, treatment of ketone **4** with AllylMgBr in the presence of CeCl$_3$ furnished a diol intermediate, which was subjected to a Pb(IV)-mediated oxidative cleavage. A follow-up NaBH$_4$ reduction yielded a somewhat unstable β,γ-unsaturated ketone **16**, which was immediately subjected to iodination producing alkyl iodide **17**. Subsequently, treating **17** with LDA triggered an intramolecular alkylation to afford α-vinyl functionalized ketones **18a** (minor, 25%) and **18b** (major, 50%). The stereoconfigurations of **18a** and **18b** were assigned at later stage, according to a single-crystal X-ray diffraction (XRD) data of compound **22**. Initially, we envisioned to subject both isomers of **18** to a Pd-catalyzed carbonylative cyclization[62,63], however preparation of the corresponding enol triflate failed. Alternatively, the terminal alkene motif in **18b** was regioselectively hydroformylated following Shi's protocol furnishing aldehyde **19** in 75% yield[64]. Any branched side products were not detected during this transformation. Although aldehyde **19** could also be synthesized via rather routine methods[33], the intrinsic structural complexity of substrate **18b** enabled the exciting opportunity for expanding the substrate scope for Shi's regioselective hydroformylation. Subsequently, an intramolecular pinacol coupling reaction mediated by SmI$_2$ furnished diol **20**, which possesses the essential 7/5 bicyclic ring scaffold. Selective acylation of the secondary hydroxyl, followed by DDQ-mediated debenzylation afforded primary alcohol **21**, which was then converted into aldehyde **22** using Dess-Martin oxidation. The identity of **22** was unambiguously confirmed via XRD analysis. A Horner–Wadsworth–Emmons homologation using phosphonate **23**[32,33,65], followed by one-pot DIBAL-H reduction afforded ketene dithioacetal **25** in 94% yield. Notably, the intermediate acetate **24** gave a single crystal suitable for XRD analysis.

At this stage, aldol-type[16,21,22] or Prins-type reaction was required to form the critical C$_{14}$–C$_{15}$ bond in **26**. However, ketene dithioacetal-involved aldol- or Prins-type reactions are poorly

explored. There are only a handful of examples reported in an intermolecular manner[54–57]. Inspired by these investigations, it was envisaged that oxidation of the hydroxyl at C-15 would produce the corresponding ketone motif, and the following ketene dithioacetal Prins-type reaction should form the crucial C–C bond thus constructing the desired pentacycle. To our delight, subjecting diol **25** to a TFAA/DMSO-mediated Swern oxidation triggered the desired cyclization of the dithioacetal ketene onto a newly formed ketone moiety. The ultimate product **26** was then formed by 1,2-addition of dimethyl sulfide onto the sulfonium intermediate, followed by demethylation[66]. While all reported examples of intermolecular ketene dithioacetal aldol/Prins-type reactions were performed under strong Lewis acid conditions such as BF$_3$•Et$_2$O, TiCl$_4$, and TMSOTf[54–57], the addition of a strong Lewis acid were not necessary for our ketone intermediate derived from **25**. Moreover, using other oxidants such as (COCl)$_2$/DMSO, Py•SO$_3$/DMSO, NCS/DMSO, or Ac$_2$O/DMSO resulted in decomposition of **25** giving poorly identifiable side products. The 2-(methylthio)-1,3-dithiane moiety in **26** was then smoothly transformed to the methyl ester **27** by the action of methanolic iodine. Selective elimination of the hydroxyl at C-15 of **27** was required to avoid any possible side reaction between the α-aminyl radical and the C$_9$–C$_{10}$ alkene moiety in a later stage of our synthesis[33]. Gratifyingly, treating cis-diol **27** with SOCl$_2$ yielded the dialkyl sulfite **28**, which upon treatment with DBU suffered E2cB elimination providing the hope-for allylic alcohol **29**. Removal of the tosyl appendage in **29** was followed by a one-pot alkylation to yield vinyl bromide **30**. This vinyl bromide substrate enabled the investigation of the Ni(0)-catalyzed C–C-coupling and the Tin-mediated radical cyclization approaches. A Ni(0)-mediated reductive coupling of **30**[53], unfortunately, failed to produce **31** but only messy results. There was not any identifiable product that could be isolated. LC-MS indicates only a trace amount of the desired hexacycle **31**. Alternatively, subjecting **30** to an AIBN/Bu$_3$SnH -mediated radical cyclization assembled the key tetrahydropyrrole ring in **31**. The acidic workup promoted the elimination of the C-9 hydroxyl group to

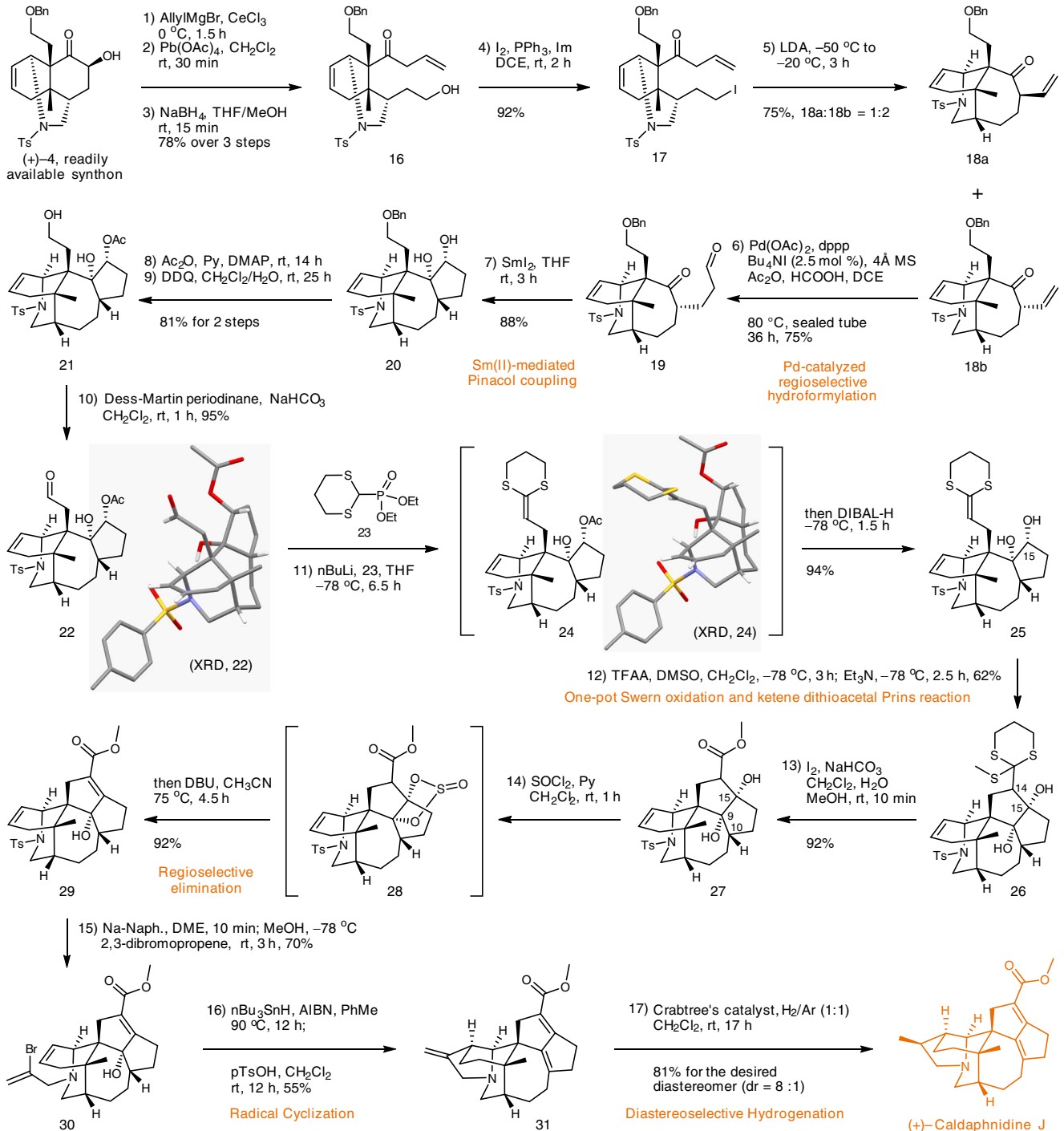

**Fig. 4 Total synthesis of (+)-caldaphnidine J.** Key transformations were highlighted in colored text. Ac acetyl, AIBN azodiisobutyronitrile, DBU 1,8-diazabicyclo[5.4.0]undec-7-ene, DCE dichloroethane, DDQ 2,3-dichloro-5,6-dicyano-1,4-benzoquinone, DIBAL-H diisobutylaluminium hydride, DMAP 4-dimethylaminopyridine, DME 1,2-dimethoxyethane, dppp 1,3-bis(diphenylphosphino) propane, Naph. naphthalene, TFAA trifluoroacetic anhydride.

yield the conjugated diene. Lastly, a highly regio- and diastereoselective hydrogenation ($H_2/Ar = 1:1$, Crabtree's catalyst) of **31** successfully produced (+)-caldaphnidine J in 81% yield (dr = 8:1). The synthetic (+)-caldaphnidine J gave spectral characteristics ($^1H$- and $^{13}C$-NMR spectroscopy and HRMS data) consistent with those of the naturally occurring (+)-caldaphnidine J, while the optical rotation is also in perfect agreement with that of the natural product (synthetic: $[\alpha]_D^{20} = +60.0$ ($c = 0.1$ in MeOH); natural: $[\alpha]_D^{20} = +57.0$ ($c = 0.2$ in MeOH)[50].

## Discussion

Owing to their extremely challenging structures, the synthesis of yuzurimine-type *Daphniphyllum* alkaloids remain unexplored for more than half a century. These intriguing alkaloids provide ideal platforms for developing and probing various synthetic strategies and methods. In this paper, we have accomplished an asymmetric synthesis of highly challenging yuzurimine-type *Daphniphyllum* alkaloid (+)-caldaphnidine J in 17 steps from readily available chiral synthon (+)−**4**. This work achieves the synthesis of a

member of the largest yet unexplored subfamily of *Daphni-phyllum* alkaloids. The highlights of our synthesis include: (1) a highly regioselective Pd-catalyzed hydroformylation reaction; (2) a Sm(II)-mediated pinacol coupling that produced a highly challenging 7/5 bicyclic system while all other attempts failed; (3) a one-pot Swern oxidation/ketene dithioacetal Prins reaction; (4) a regioselective elimination through a cyclic sulfite intermediate, and (5) a radical cyclization reaction that rapidly constructed the tetrahydropyrrole motif. The synthetic strategies and methods should inspire further advances in the synthesis of diverse *Daphniphyllum* alkaloids and related natural products.

## Methods

**General**. Unless indicated, all commercially available reagents and anhydrous solvents were purchased at the highest commercial quality and were used as received without further purification. All non-aqueous reactions were carried out under argon atmosphere using dry glassware that had been flame-dried under a stream of argon unless otherwise noted. Tetrahydrofuran (THF) was distilled from sodium benzophenone under argon atmosphere. Dichloromethane (DCM) was distilled from calcium hydride. Reactions were monitored by thin-layer chromatography (TLC; GF254) using plates supplied by Yantai Chemicals (China) and visualized under UV or by staining with an ethanolic solution of phosphomolybdic acid, cerium sulfate or iodine. Flash column chromatography was performed using silica gel (particle size, 0.040–0.063 mm).

NMR spectra were recorded on a Bruker AV400 MHz or a Bruker AscendTM 500 MHz instrument and calibrated using residual undeuterated chloroform, methanol, and dichloromethane in $CDCl_3$ ($\delta$ H = 7.26 ppm, $\delta$ C = 77.0 ppm), MeOD-d4 ($\delta$ H = 3.31 ppm, $\delta$ C = 49.0 ppm), or $CD_2Cl_2$ ($\delta$ H = 5.32 ppm, $\delta$ C = 53.84 ppm) as an internal reference. The following abbreviations were used to describe signal multiplicities: s, singlet; d, doublet; t, triplet; dt, double triplet; ddd, doublet of double doublet; ddt, doublet of double triplet; m, multiplet. High-resolution mass spectra (HRMS) were recorded on a Thermo Scientific Q Exactive Hybrid Quadrupole Orbitrap mass spectrometer.

**Experimental data**. For NMR spectra of synthetic intermediates, see Supplementary Figs. 3–57. For the experimental procedures and spectroscopic and physical data of compounds and the crystallographic data of compound **22** and **24**, see Supplementary Methods.

**Supplementary Information** accompanies this manuscript.

## Data availability

The X-ray crystallographic coordinates for structures **22** and **24** reported in this study have been deposited at the Cambridge Crystallographic Data Centre (CCDC) with the accession codes CCDC 1986889 and CCDC 1986960 (www.ccdc.cam.ac.uk/data_request/cif), respectively. We declare that all other relevant data supporting the findings of this study are available within the article and its Supplementary Information files.

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

## Acknowledgements

Financial support from Shenzhen Science and Technology Innovation Committee (JCYJ20170817110515599, KQJSCX20170728154233200, and KQTD20150717103157174), NSFC (21772082 and 21971104), Shenzhen Nobel Prize Scientists Laboratory Project (C17783101), Guangdong Innovative Program (No. 2019BT02Y335) and Guangdong Provincial Key Laboratory of Catalysis (No. 2020B121201002) are greatly appreciated. We also thank SUSTech CRF NMR facility, Dr. Xiaoyong Chang (SUSTech) for XRD analysis and Dr. Yang Yu (SUSTech) for HRMS analysis.

## Author contributions

J.X. conceived and designed the project and wrote the paper with assistance from L.-D.G., Y.Z., and J.H.; L.-D.G., Y.Z, J.H., C.N., H.F., and Y.C. performed the experiments. All authors discussed the results and commented on the manuscript. Y.Z. and J.H. contributed equally to this research.

## Competing interests

The authors declare no competing interests.
