## [Peer Review File · Nature Communications]

REVIEWER COMMENTS

Reviewer #1 (Remarks to the Author):

This manuscript describes the first total synthesis of a yuzurimine-type daphniphyllum alkaloid (+)-caldaphnidine J, which is another synthesis of the daphniphyllum alkaloids that this group has had great achievements on for the past few years. Congratulations to the team for completing yet another one of these complex molecules!

The synthetic route itself has much basis in the previous chemistry reported by this group. The proposal of constructing the 7/5 system using Saa's methodology is quite novel despite being unsuccessful in the actual synthesis.

The successful route includes three major strategies: enolate alkylation to form the 7-membered system; formation of the 5-membered system; aldol-type chemistry for the final ring. The sequence is influenced by the Smith synthesis of (-)-calcyphylline A as noted by the authors, which somewhat lowers the overall originality of the work. Apart from synthesizing different classes of daphniphyllum alkaloids, the differences are the formation of the second 5-membered system (hydroformylation/SmI2 coupling compared to Nazarov cyclization) and a different method for finishing the net ester-aldol reaction for the construction of the final ring.

This final ring closure is the largest highlight of this sequence. First, a one-pot oxidation/C-C bond formation is very effective, and second, the use of vinyl dithiane as an ester enolate equivalent remains somewhat unexplored. It is interesting that the TFAA/DMSO protocol selectively activates the hydroxyl group in the presence of the electron rich vinyl dithiane. The use of this rather new chemistry in such a complex setting is especially noteworthy. This step itself and the accomplishment of the synthesis justify this article's publication in Nature Communications.

The supporting information is well-prepared with clear spectra, some minor notes on the written section:

Compound 5 contains 35 hydrogen, NMR data showed 37

Compound SI reported 32 carbon for a structure with 33, what is also missing is the C-F coupling of the triflate group.

An extra carbon seemed to be counted in the ¹³CNMR of 3

Compound 9 contains 47 hydrogen yet the data counted for a total of 50. The authors wrote 1.18-1.13 (m, 4H) and 1.12 (s, 3H), whereas according to the spectra which is fine, the 4H integrate at this area included the singlet. This would account for the extra hydrogen.

Compound 11 contains 33 hydrogen yet the data counted for a total of 34

Compound 16: the solvent peaks were also typed in the experimental section, which should be removed.

Compound 17 contains 38 hydrogens yet the data counted for a total of 42. The spectra is fine, but the data written for the number of hydrogen in the experimental section were mistyped for ppm>4

The HRMS for compound 18a should be [H37] instead of [H38]. The same also applies for the diastereomeric 18b, the authors typed [H39] instead of [H37].

There seems to be missing hydrogen in the data for 21 and 22, which may be attributed to the

hydroxyl protons. The hydrogen written for compound 22 does not seem to match the spectra integrates.

The integrate of ¹H NMR for compound 25 at 1.93-1.69 should be 4 instead of 5.

Compound 26: the typed data of ¹³C NMR in the experimental section contained the 3 solvent peaks which should be removed (similar problem occurred with 16 (vide supra)).

Reviewer #2 (Remarks to the Author):

This is a beautifully executed synthetic study on total synthesis of (+)-caldaphnidine J. The complexity of the target and elegance of the synthetic route render the current manuscript of significant interest to the readership of Nature Communications.

However, the authors' remarkable accomplishment is described in a manner that hardly be regarded as elegant. The manuscript is written very poorly; it was the absolute struggle to read and follow the science. With improving the readability of the manuscript, this outstanding piece of work is enthusiastically recommended for acceptance in Nature Communications.

Respectfully,
Daler Baidilov

The following are edits and suggestions to improve the readability of the manuscript:

Page 1, line 9
remove "and"
...the first enantioselective total synthesis...

Page 1, line 11
...is described...

Page 1, line 12
remove "the all other attempts" part of the sentence
... Pd-catalyzed hydroformylation, samarium(II)-mediated pinacol coupling, and a one-pot Swern oxidation/ketene dithioacetal Prins reaction.

Page 2, line 27 Remove comma
...complex caged...

Page 2, line 29
I would bracket biological profile examples and reword the follow-up statement:
... biological profiles (such as anticarcinogenic, neurotrophic and anti-HIV activity), these alkaloids have drawn much attention from the synthetic community.

Page 2, line 32
Remove the list of subfamilies. These will disrupt the reader. The same for the follow-up sentence.

Page 2, line 40
...total synthesis of yuzurimine-type alkaloids...

Page 2, line 42
...Since the milestone achievement by Hirata in 1966, nearly fifty...

Page 2, line 43
...which account for...

Page 2, line 45
Remove the sentence starting on this line. This information is already given in Figure 1.

Page 2, line 47
Please simplify the sentence
...Caldaphnidine J, our target molecule, was isolated by the Yue group in 2008. It possesses a hexacyclic scaffold...

Page 3, line 49
... carboxylic ester thus signifying a formidable...

Page 3, line 50
...long-lasting interest in...

Page 3, line 51
...we now wish to report the first asymmetric total synthesis of yuzurimine-type alkaloid (+)-caldaphnidine J.

Page 3, caption for Figure 1
You have already made this statement, please remove this sentence (starting on line 55).
"Our target molecule, caldaphnidine J, are highlighted in colour, which posed a formidable synthetic challenge for accessing its total synthesis"

Page 3, line 59
...analysis indicates...

Page 3, line 60
...via Ni(0)-mediated...

Page 4, caption for Figure 2
The analysis has already been summarized in the main text, please remove.

Page 4, line 69
...investigation focused...

Page 4, line 70
The treatment of the known tricycle 4 with MeMgBr in the presence of CeCl₃ was followed by an oxidative cleavage of the intermediate diol producing dicarbonyl 5.

Page 4, line 73
...Following Stang's protocol, ketone moiety in 6 was converted into an enol triflate, which was then treated with pyridine affording alkyne 7.

Page 4, line 75
Inspired by Saá's impressive Nazarov carbocyclization cascade, enyne acetal 3 was treated with Brønsted acids such as TFA and HBF₄·OEt₂. However, none of the desired hydroazulenone 8 was detected.

Page 4, line 78
...formic acid-promoted...

Page 4, line 79

...between the thioalkyne and the acetal motifs in compound 9 returned only messy mixtures.

Page 4, line 80

... cyclization was prevented by the steric bulk at C-8 (caldaphnidine J numbering)... Include the numbering in the scheme as well. This will help a lot in following what is written in the manuscript.

Page 5, caption for Figure 3

Remove the summary

Page 5, line 88

... aldol condensation of 5 yielded solely cyclopentene 11, but not cycloheptenone 12 possessing the desired 7-membered ring.

It is exceedingly difficult to follow the text that consists of "compound A" and "product B". Please provide a description to your intermediates.

Page 5, line 90

Use either "Appel reaction" or "iodination"

Page 5, line 91

... upon treatment with LDA yielding the desired tricycle 13 in 69% overall yield.

Page 5, line 92

...it was envisioned...

Page 5, line 93

...Nazarov cyclization would elaborate the desired bicycle 15...

Page 5, line 94

While ketone 13 is quite similar to Smith's substrates, in our hands all of the deprotonation attempts failed resulting in no reaction at negative and decomposition events at elevated temperatures. It is quite possible that the corresponding enol triflate 14 (and hence the free enol form) is unstable introducing increased ring strain to a 6/6/7 tricyclic system.

Page 6, line 95

Remove "initial"

Page 6, line 101

As depicted in Fig. 4, treatment of ketone 4 with AllylMgBr in the presence of CeCl_3 furnished a diol intermediate, which was subjected to a Pb(IV) -mediated oxidative cleavage. A follow-up NaBH_4 reduction yielded...

Page 6, line 103

...subjected to iodination producing alkyl iodide 17.

Page 6, line 106

...at later stage...

...according to a single crystal...

Page 6, line 107

Reword the sentence starting on this line as follows:

Initially, we envisioned to subject both isomers of 18 to a Pd-catalyzed carbonylative cyclization, however preparation of the corresponding enol triflate failed. Alternatively, the terminal alkene motif in 18b was selectively hydroformylated following Shi's protocol furnishing 19 in 75% yield. Any branched side products were not detected during this transformation.

Page 6, line 114

...furnished diol 20, which possesses...

Page 6, line 115

...the secondary hydroxyl, followed by DDQ-mediated debenzoylation afforded primary alcohol 21...

Page 6, line 117 Reword the sentence starting on this line as follows:

A Horner-Wadsworth-Emmons homologation using phosphonate 23, followed by one-pot DIBAL-H reduction afforded ketene dithioacetal 25 in 94% yield. Notably, the intermediate acetate 24 gave a single crystal suitable for XRD analysis.

Page 6, line 121

At this stage, aldol-type...

Page 6, line 122

Remove sentence starting on this line. This is a common sense.

Page 6, line 124

However, ketene dithioacetal-involved aldol- or Prins-type reactions are poorly explored. There are only handful of examples reported in the literature.

Page 7, line 126

...oxidation of the hydroxyl at C-15...

Page 7, line 128

...the crucial C-C bond thus constructing the desired pentacycle.

Page 7, line 129

To our delight, subjecting diol 25 to a TFAA/DMSO-mediated Swern oxidation triggered a 5-exo-trig cyclization of the dithioacetal ketene onto a newly formed ketone moiety. The ultimate product 26 was then formed by 1,2-addition of dimethyl sulfide onto the sulfonium intermediate, followed by demethylation.

Page 7, line 133

While all the reported...

Page 7, line 134

...were performed...

Page 7, line 135

...TMSOTf, addition of a strong Lewis acid was not unnecessary for a ketone intermediate derived from 25.

Page 7, line 136

Moreover, using other oxidants such as (COCl)₂/DMSO, Py•SO₃/DMSO, NCS/DMSO, and Ac₂O/DMSO resulted in either decomposition of 25 or a messy reaction mixture.

To what does 25 decompose to? I, as a reader, would be very curious to find out. Please describe the decomposition product.

Page 7, line 138

I wouldn't call the transformation of 26 to 27 as oxidative. The formal oxidation state of the carbon stays intact.

I would reword it as such: "The 2-(methylthio)-1,3-dithiane moiety in 26 was then smoothly transformed to methyl ester 27 by the action of a methanolic iodine."

Page 7, line 139

To avoid any possible side reaction between...
...in a later stage...

Page 7, line 141

There is no need in mentioning the C-9 hydroxy, as you already used the word "selective"!
...our synthesis, a selective elimination of the hydroxyl at C-15 of 27 was required.

Page 7, line 143

...sulfite 28, which upon treatment with DBU suffered E2cB-elimination providing the hoped-for allylic alcohol 29.

Page 7, line 145

Describe your intermediates to a reader!

Removal of the tosyl appendage in 29 was followed by a one-pot alkylation to yield vinyl bromide 30.

Page 7, line 145

This is the second time you use "messy results". Please, inform the reader what went wrong in this transformation and what was the major side-product.

Page 7, line 147

...to a AIBN/Bu₃SnH-mediated...

Page 7, line 149

Lastly, a highly regio- and diastereoselective hydrogenation...

Page 9, line 165

This is rather a summary, not discussion. Please change.

Page 9, line 166

... the first asymmetric synthesis...
... in 17 steps from... (remove comma)

Page 9, line 167

This work achieved the first synthesized...

Page 9, line 170

...coupling that produced a highly challenging 7/5 bicyclic system while all other attempts failed.

Page 9, line 172

...through a cyclic sulfite...

Page 9, line 173

... that rapidly constructed...

References

Please correct all the titles for the racemic syntheses: appear as (.+-.)

Some of the references contain "et al". Please acknowledge all the authors.

Reviewer #3 (Remarks to the Author):

Xu and co-workers report the first total synthesis of (+)-caldaphnidine J, a complex yuzurimine-type alkaloid. The synthesis of such highly condensed polycyclic compounds generally poses challenges in construction of rings onto polycyclic intermediate. The authors successfully use a samarium(II)-mediated pinacol coupling and a ketene dithioacetal Prins reaction for such ring construction. Considering the significance of this study for synthetic community, this reviewer recommends publishing this study in *Nat. Commun.* However, this manuscript does not describe the novelty of this work accurately as shown below:

Ref. 33, in which Xu's total synthesis of a related alkaloid is described, includes a lot of strategic similarity to the synthesis in this manuscript. The authors should add explanation of the similarity to the first or second sentence of p6 (the part of "Total synthesis of (+)-caldaphnidine J"). For example, "The fruitless initial synthetic attempts toward the 7/5 bicyclic ring scaffold forced us to use the similar strategy to the total synthesis of caldaphnidine O".

To construct the cycloheptane ring, the same strategy (S_N2 reaction) is used. The difference between this manuscript and ref. 33 is only the number of methylene, which comes from the Grignard reagents (allyl one in this manuscript and 3-buten-1-yl one in ref. 33). The key ketoaldehyde 19 is synthesized by Pd-catalyzed regioselective hydroformylation from vinylcycloheptanone 18b in this manuscript. On the other hand, the corresponding borane was prepared from allylcycloheptanone 13a in ref. 33, which should be converted to ketoaldehyde 19 in this manuscript via conventional oxidation steps. The readers should wonder why the authors set the little different intermediate. Please add explanation about the similarity and the difference between the synthesis of ref. 33 and that of this manuscript.

Similar radical cyclization for the last ring construction (30 to 31 in this manuscript and 5 to 16 in ref. 33) is used. The difference is the structure of the substituents (a 2-bromoallyl group in this manuscript and a propargyl group in ref. 33). Is the difference necessary? Please describe the effect of the substituent if they examined a propargyl group for this synthesis.

This reviewer hopes that this manuscript is published after the above-mentioned points are described accurately.

Dear Reviewers,

We greatly appreciate the supports, suggestions and comments from your side. Enclosed please find our revised manuscript and the point-to-point responses. We have very carefully revised our manuscript by strictly following your suggestions and comments. Moreover, we have shown all changes in the manuscript text file with track changes.

As shown in below, Reviewer's original comments are *Italic* and our responses are highlighted in blue.

Point-to-point response to Reviewer 1:

This manuscript describes the first total synthesis of a yuzurimine-type daphniphyllum alkaloid (+)-caldaphnidine J, which is another synthesis of the daphniphyllum alkaloids that this group has had great achievements on for the past few years. Congratulations to the team for completing yet another one of these complex molecules!

The synthetic route itself has much basis in the previous chemistry reported by this group. The proposal of constructing the 7/5 system using Saa's methodology is quite novel despite being unsuccessful in the actual synthesis.

The successful route includes three major strategies: enolate alkylation to form the 7-membered system; formation of the 5-membered system; aldol-type chemistry for the final ring. The sequence is influenced by the Smith synthesis of (-)-calcyphylline A as noted by the authors, which somewhat lowers the overall originality of the work. Apart from synthesizing different classes of daphniphyllum alkaloids, the differences are the formation of the second 5-membered system (hydroformylation/SmI2 coupling compared to Nazarov cyclization) and a different method for finishing the net ester-aldol reaction for the construction of the final ring.

This final ring closure is the largest highlight of this sequence. First, a one-pot oxidation/C-C bond formation is very effective, and second, the use of vinyl dithiane as an ester enolate equivalent remains somewhat unexplored. It is interesting that the TFAA/DMSO protocol selectively activates the hydroxyl group in the presence of the electron rich vinyl dithiane. The use of this rather new chemistry in such a complex setting is especially noteworthy. This step itself and the accomplishment of the synthesis justify this article's publication in Nature Communications."

We greatly appreciate these kind comments. No revision suggested for the manuscript text.

The supporting information is well-prepared with clear spectra, some minor notes on the written section:

Compound 5 contains 35 hydrogen, NMR data showed 37

We are highly grateful to reviewer's very detailed comments. We have re-integrated the ¹H-NMR by following the above comments.

Compound SI reported 32 carbon for a structure with 33, what is also missing is the C-F coupling of the triflate group.

We have recollected ¹³C-NMR for this compound. We have also collected a ¹⁹F-NMR for this compound, please see enclosed revised SI.

An extra carbon seemed to be counted in the ¹³CNMR of 3

We have corrected the peak picking in the ¹³C-NMR by following the above comments.

Compound 9 contains 47 hydrogen yet the data counted for a total of 50. The authors wrote 1.18-1.13 (m, 4H) and 1.12 (s, 3H), whereas according to the spectra which is fine, the 4H integrate at this area included the singlet. This would account for the extra hydrogen.

We have re-integrated the ¹H-NMR by following the above comments.

Compound 11 contains 33 hydrogen yet the data counted for a total of 34

We have re-integrated the ¹H-NMR by following the above comments.

Compound 16: the solvent peaks were also typed in the experimental section, which should be removed.

We have removed the solvent peak data accordingly.

Compound 17 contains 38 hydrogens yet the data counted for a total of 42. The spectra is fine, but the data written for the number of hydrogen in the experimental section were mistyped for ppm>4

We have corrected the hydrogen counting.

The HRMS for compound 18a should be [H37] instead of [H38]. The same also applies for the diastereomeric 18b, the authors typed [H39] instead of [H37].

We have corrected the HRMS for these compounds.

There seems to be missing hydrogen in the data for 21 and 22, which may be attributed to the hydroxyl protons. The hydrogen written for compound 22 does not seem to match the spectra integrates.

We fully agree with this reviewer that the missing hydrogen should be attributed to the OH proton. We have also corrected the hydrogen counting for compound **22**.

The integrate of 1HNMR for compound 25 at 1.93-1.69 should be 4 instead of 5.

We have re-integrated the ¹H-NMR by following the above comments.

Compound 26: the typed data of 13CNMR in the experimental section contained the 3 solvent peaks which should be removed (similar problem occurred with 16 (vide supra).

We have revised the ¹³C-NMR data accordingly.

Point-to-point response to Reviewer 2:

This is a beautifully executed synthetic study on total synthesis of (+)-caldaphnidine J. The complexity of the target and elegance of the synthetic route render the current manuscript of significant interest to the readership of Nature Communications.

However, the authors' remarkable accomplishment is described in a manner that hardly be regarded as elegant. The manuscript is written very poorly; it was the absolute struggle to read and follow the science. With improving the readability of the manuscript, this outstanding piece of work is enthusiastically recommended for acceptance in Nature Communications.

The following are edits and suggestions to improve the readability of the manuscript:

We greatly appreciate these kind comments. We have thoroughly revised our manuscript text by strictly following these suggestions.

Page 1, line 9, remove "and"

...the first enantioselective total synthesis..."

We have revised our manuscript text accordingly.

Page 1, line 11

...is described...

We have revised our manuscript text accordingly.

Page 1, line 12

remove "the all other attempts" part of the sentence

... Pd-catalyzed hydroformylation, samarium(II)-mediated pinacol coupling, and a one-pot Swern oxidation/ketene dithioacetal Prins reaction.

We have revised our manuscript text accordingly.

Page 2, line 27 Remove comma

...complex caged...

We have revised our manuscript text accordingly.

Page 2, line 29

I would bracket biological profile examples and reword the follow-up statement:

... biological profiles (such as anticarcinogenic, neurotrophic and anti-HIV activity), these alkaloids have drawn much attention from the synthetic community.

We have revised our manuscript text accordingly.

Page 2, line 32

Remove the list of subfamilies. These will disrupt the reader. The same for the follow-up sentence.

We have revised our manuscript text accordingly.

Page 2, line 40

...total synthesis of yuzurimine-type alkaloids...

We have revised our manuscript text accordingly.

Page 2, line 42

...Since the milestone achievement by Hirata in 1966, nearly fifty...

We have revised our manuscript text accordingly.

Page 2, line 43

...which account for...

We have revised our manuscript text accordingly.

Page 2, line 45

Remove the sentence starting on this line. This information is already given in Figure 1.

We have revised our manuscript text accordingly.

Page 2, line 47

Please simplify the sentence

...Caldaphnidine J, our target molecule, was isolated by the Yue group in 2008. It possesses a hexacyclic scaffold....

We have revised our manuscript text accordingly.

Page 3, line 49

... carboxylic ester thus signifying a formidable...

We have revised our manuscript text accordingly.

Page 3, line 50

...long-lasting interest in...

We have revised our manuscript text accordingly.

Page 3, line 51

...we now wish to report the first asymmetric total synthesis of yuzurimine-type alkaloid (+)-caldaphnidine J.

We have revised our manuscript text accordingly.

Page 3, caption for Figure 1

You have already made this statement, please remove this sentence (starting on line 55).

“Our target molecule, caldaphnidine J, are highlighted in colour, which posed a formidable synthetic challenge for accessing its total synthesis”

We have revised our manuscript text accordingly.

Page 3, line 59

...analysis indicates...

We have revised our manuscript text accordingly.

Page 3, line 60

...via Ni(0)-mediated...

We have revised our manuscript text accordingly.

Page 4, caption for Figure 2

The analysis has already been summarized in the main text, please remove.

We have revised our manuscript text accordingly.

Page 4, line 69

...investigation focused...

We have revised our manuscript text accordingly.

Page 4, line 70

The treatment of the known tricycle 4 with MeMgBr in the presence of CeCl₃ was followed by an oxidative cleavage of the intermediate diol producing dicarbonyl 5.

We have revised our manuscript text accordingly.

Page 4, line 73

...Following Stang’s protocol, ketone moiety in 6 was converted into an enol triflate, which was then treated with pyridine affording alkyne 7.

We have revised our manuscript text accordingly.

Page 4, line 75

Inspired by Saá’s impressive Nazarov carbocyclization cascade, enyne acetal 3 was treated with Brønsted acids such as TFA and HBF₄·OEt₂. However, none of the desired hydroazulenone 8 was detected.

We have revised our manuscript text accordingly.

Page 4, line 78

...formic acid-promoted...

We have revised our manuscript text accordingly.

Page 4, line 79

...between the thioalkyne and the acetal motifs in compound 9 returned only messy mixtures.

We have revised our manuscript text accordingly.

Page 4, line 80

... cyclization was prevented by the steric bulk at C-8 (caldaphnidine J numbering)... Include the numbering in the scheme as well. This will help a lot in following what is written in the manuscript.

We have revised our manuscript text and scheme accordingly.

Page 5, caption for Figure 3

Remove the summary

We have revised our manuscript text accordingly.

Page 5, line 88

... aldol condensation of 5 yielded solely cyclopentene 11, but not cycloheptenone 12 possessing the desired 7-membered ring.

It is exceedingly difficult to follow the text that consists of "compound A" and "product B". Please provide a description to your intermediates.

We have revised our manuscript text accordingly.

Page 5, line 90

Use either "Appel reaction" or "iodination"

We have revised our manuscript text accordingly.

Page 5, line 91

... upon treatment with LDA yielding the desired tricycle 13 in 69% overall yield.

We have revised our manuscript text accordingly.

Page 5, line 92

...it was envisioned...

We have revised our manuscript text accordingly.

Page 5, line 93

...Nazarov cyclization would elaborate the desired bicycle 15...

We have revised our manuscript text accordingly.

Page 5, line 94

While ketone 13 is quite similar to Smith's substrates, in our hands all of the deprotonation attempts failed resulting in no reaction at negative and decomposition events at elevated temperatures. It is quite possible that the corresponding enol triflate 14 (and hence the free enol form) is unstable introducing increased ring strain to a 6/6/7 tricyclic system.

We have revised our manuscript text accordingly.

Page 6, line 95

Remove “initial”

We have revised this sentence according to Reviewer 2 and Reviewer 3’s comments.

Page 6, line 101

As depicted in Fig. 4, treatment of ketone 4 with AllylMgBr in the presence of CeCl₃ furnished a diol intermediate, which was subjected to a Pb(IV)-mediated oxidative cleavage. A follow-up NaBH₄ reduction yielded...

We have revised our manuscript text accordingly.

Page 6, line 103

...subjected to iodination producing alkyl iodide 17.

We have revised our manuscript text accordingly.

Page 6, line 106

...at later stage...

...according to a single crystal...

We have revised our manuscript text accordingly.

Page 6, line 107

Reword the sentence starting on this line as follows:

Initially, we envisioned to subject both isomers of 18 to a Pd-catalyzed carbonylative cyclization, however preparation of the corresponding enol triflate failed. Alternatively, the terminal alkene motif in 18b was selectively hydroformylated following Shi’s protocol furnishing 19 in 75% yield. Any branched side products were not detected during this transformation.

We have revised our manuscript text accordingly.

Page 6, line 114

...furnished diol 20, which possesses...

We have revised our manuscript text accordingly.

Page 6, line 115

...the secondary hydroxyl, followed by DDQ-mediated debenzylation afforded primary alcohol 21...

We have revised our manuscript text accordingly.

Page 6, line 117 *Reword the sentence starting on this line as follows:*

A Horner-Wadsworth-Emmons homologation using phosphonate 23, followed by one-pot DIBAL-H reduction afforded ketene dithioacetal 25 in 94% yield. Notably, the intermediate acetate 24 gave a single crystal suitable for XRD analysis.

We have revised our manuscript text accordingly.

Page 6, line 121

At this stage, aldol-type...

We have revised our manuscript text accordingly.

Page 6, line 122

Remove sentence starting on this line. This is a common sense.

We have revised our manuscript text accordingly.

Page 6, line 124

However, ketene dithioacetal-involved aldol- or Prins-type reactions are poorly explored. There are only handful of examples reported in the literature.

We have revised our manuscript text accordingly.

Page 7, line 126

...oxidation of the hydroxyl at C-15...

We have revised our manuscript text accordingly.

Page 7, line 128

...the crucial C-C bond thus constructing the desired pentacycle.

We have revised our manuscript text accordingly.

Page 7, line 129

To our delight, subjecting diol 25 to a TFAA/DMSO-mediated Swern oxidation triggered a 5-exo-trig cyclization of the dithioacetal ketene onto a newly formed ketone moiety. The ultimate product 26 was then formed by 1,2-addition of dimethyl sulfide onto the sulfonium intermediate, followed by demethylation.

We have revised our manuscript text accordingly.

Page 7, line 133

While all the reported...

We have revised our manuscript text accordingly.

Page 7, line 134

...were performed...

We have revised our manuscript text accordingly.

Page 7, line 135

...TMSOTf, addition of a strong Lewis acid was not unnecessary for a ketone intermediate derived from 25.

We have revised our manuscript text accordingly.

Page 7, line 136

Moreover, using other oxidants such as (COCl)₂/DMSO, Py•SO₃/DMSO, NCS/DMSO, and Ac₂O/DMSO resulted in either decomposition of 25 or a messy reaction mixture. To what does 25 decompose to? I, as a reader, would be very curious to find out. Please describe the decomposition product.

We are very thankful to these comments. Actually, we have tried to find out what were happened during the decompositions. However, extremely messy results from the decomposition prevented us to purify or identify the side products. We have revised our manuscript text accordingly by revise the sentence as “Moreover, using other oxidants such as (COCl)₂/DMSO, Py•SO₃/DMSO, NCS/DMSO, or Ac₂O/DMSO resulted in decomposition of **25** giving poorly identifiable side products.”

Page 7, line 138

I wouldn't call the transformation of 26 to 27 as oxidative. The formal oxidation state of the carbon stays intact.

I would reword it as such: “The 2-(methylthio)-1,3-dithiane moiety in 26 was then smoothly transformed to methyl ester 27 by the action of a methanolic iodine.”

We have revised our manuscript text accordingly.

Page 7, line 139

To avoid any possible side reaction between...

...in a later stage...

We have revised our manuscript text accordingly.

Page 7, line 141

There is no need in mentioning the C-9 hydroxy, as you already used the word “selective”!

...our synthesis,³³ a selective elimination of the hydroxyl at C-15 of 27 was required.

We have revised our manuscript text accordingly.

Page 7, line 143

...sulfite 28, which upon treatment with DBU suffered E2cB-elimination providing the hoped-for allylic alcohol 29.

We have revised our manuscript text accordingly.

Page 7, line 145

Describe your intermediates to a reader!

Removal of the tosyl appendage in 29 was followed by a one-pot alkylation to yield vinyl bromide 30.

We have revised our manuscript text accordingly.

Page 7, line 145

This is the second time you use “messy results”. Please, inform the reader what went wrong in this transformation and what was the major side-product.

We much appreciate these comments. We have tried to find out what were happened during the decompositions. However, extremely messy results from the decomposition prevented us to purify or identify the side products. We have added sentences as “There was not any identifiable product that could be isolated. LC-MS indicates only a trace amount of the desired hexacycle **31**.”

Page 7, line 147

...to a AIBN/Bu₃SnH-mediated...

We have revised our manuscript text accordingly.

Page 7, line 149

Lastly, a highly regio- and diastereoselective hydrogenation...

We have revised our manuscript text accordingly.

Page 9, line 165

This is rather a summary, not discussion. Please change.

We have revised the Discussion Section accordingly, by adding discussions as “Owing to their extremely challenging structures, the synthesis of yuzurimine-type *Daphniphyllum* alkaloids remain unexplored for more than half a century. These intriguing alkaloids provide ideal platforms for developing and probing novel strategies and synthetic methods. In this paper, ...”

Page 9, line 166

... the first asymmetric synthesis...

... in 17 steps from... (remove comma)

We have revised our manuscript text accordingly.

Page 9, line 167

This work achieved the first synthesized...

We have revised our manuscript text accordingly.

Page 9, line 170

...coupling that produced a highly challenging 7/5 bicyclic system while all other attempts failed.

We have revised our manuscript text accordingly.

Page 9, line 172

...through a cyclic sulfite...

We have revised our manuscript text accordingly.

Page 9, line 173

... that rapidly constructed...

We have revised our manuscript text accordingly.

References Please correct all the titles for the racemic syntheses: appear as (.+-.)

Some of the references contain “et al”. Please acknowledge all the authors.

We have revised the reference section accordingly. The author names are listed following the *Nature Communications* Author Guide: “All authors should be included in reference lists unless there are six or more, in which case only the first author should be given, followed by 'et al.'”

Point-to-point response to Reviewer 3:

Xu and co-workers report the first total synthesis of (+)-caldaphnidine J, a complex yuzurimine-type alkaloid. The synthesis of such highly condensed polycyclic compounds generally poses challenges in construction of rings onto polycyclic intermediate. The authors successfully use a samarium(II)-mediated pinacol coupling and a ketene dithioacetal Prins reaction for such ring construction. Considering the significance of this study for synthetic community, this reviewer recommends publishing this study in Nat. Commun.

We greatly appreciate these kind comments.

However, this manuscript does not describe the novelty of this work accurately as shown below:

Ref. 33, in which Xu's total synthesis of a related alkaloid is described, includes a lot of strategic similarity to the synthesis in this manuscript. The authors should add explanation of the similarity to the first or second sentence of p6 (the part of "Total synthesis of (+)-caldaphnidine J"). For example, "The fruitless initial synthetic attempts toward the 7/5 bicyclic ring scaffold forced us to use the similar strategy to the total synthesis of caldaphnidine O".

We have revised our manuscript text accordingly, by adding the sentence suggested.

To construct the cycloheptane ring, the same strategy (SN2 reaction) is used. The difference between this manuscript and ref. 33 is only the number of methylene, which comes from the Grignard reagents (allyl one in this manuscript and 3-buten-1-yl one in ref. 33). The key ketoaldehyde 19 is synthesized by Pd-catalyzed regioselective hydroformylation from vinylcycloheptanone 18b in this manuscript. On the other hand, the corresponding borane was prepared from allylcycloheptanone 13a in ref. 33, which should be converted to ketoaldehyde 19 in this manuscript via conventional oxidation steps. The readers should wonder why the authors set the little different intermediate. Please add explanation about the similarity and the difference between the synthesis of ref. 33 and that of this manuscript.

We greatly appreciate these comments. We have addressed these concerns by adding a sentence in the manuscript text as "Although aldehyde **19** could also be synthesized via rather routine methods,³³ the intrinsic structural complexity of substrate **18b** enabled the exciting opportunity for expanding the substrate scope for Shi's regioselective hydroformylation."

Similar radical cyclization for the last ring construction (30 to 31 in this manuscript and 5 to 16 in ref. 33) is used. The difference is the structure of the substituents (a 2-bromoallyl group in this manuscript and a propargyl group in ref. 33) Is the difference

necessary? Please describe the effect of the substituent if they examined a propargyl group for this synthesis.

This reviewer hopes that this manuscript is published after the above-mentioned points are described accurately.

As originally designed, we were planning to use the vinyl bromide substrate to investigate the Ni(0)-mediated C-C coupling reaction. Although the desired reductive coupling was unsuccessful, the vinyl bromide was then used for the radical cyclization approach. Therefore, we did not prepare the corresponding *N*-propargyl analogue, although it is believed that this analogue should work also well in the same radical reaction conditions. We have added explanation in our manuscript text as “This vinyl bromide substrate enabled the investigation of the Ni(0)-catalyzed C-C-coupling and the Tin-mediated radical cyclization approaches.”

After carefully made the revision and reformatting, we believe that our newly revised manuscript should now met the publication criteria for *Nature Communications*.

Again, thank you very much!